# Hepatitis C Virus Down-Regulates the Expression of Ribonucleotide Reductases to Promote Its Replication

**DOI:** 10.3390/pathogens12070892

**Published:** 2023-06-29

**Authors:** Chee-Hing Yang, Cheng-Hao Wu, Shih-Yen Lo, Ahai-Chang Lua, Yu-Ru Chan, Hui-Chun Li

**Affiliations:** 1Department of Microbiology and Immunology, School of Medicine, Tzu Chi University, Hualien 97004, Taiwan; cheehing2@gms.tcu.edu.tw; 2Department of Laboratory Medicine, Buddhist Tzu Chi General Hospital, Hualien 97004, Taiwan; hao9981@tzuchi.com.tw (C.-H.W.); losylo@mail.tcu.edu.tw (S.-Y.L.); 3Department of Laboratory Medicine and Biotechnology, School of Medicine, Tzu Chi University, Hualien 97004, Taiwan; ahai@mail.tcu.edu.tw (A.-C.L.); microbiology721@gms.tcu.edu.tw (Y.-R.C.); 4Department of Biochemistry, School of Medicine, Tzu Chi University, Hualien 97004, Taiwan

**Keywords:** hepatitis C virus, ribonucleotide reductase, NTP/dNTP ratio, Didox, Trimidox, hydroxyurea, NS5A

## Abstract

Ribonucleotide reductases (RRs or RNRs) catalyze the reduction of the OH group on the 2nd carbon of ribose, reducing four ribonucleotides (NTPs) to the corresponding deoxyribonucleotides (dNTPs) to promote DNA synthesis. Large DNA viruses, such as herpesviruses and poxviruses, could benefit their replication through increasing dNTPs via expression of viral RRs. Little is known regarding the relationship between cellular RRs and RNA viruses. Mammalian RRs contain two subunits of ribonucleotide reductase M1 polypeptide (RRM1) and two subunits of ribonucleotide reductase M2 polypeptide (RRM2). In this study, expression of cellular RRMs, including RRM1 and RRM2, is found to be down-regulated in hepatitis C virus (HCV)-infected Huh7.5 cells and Huh7 cells with HCV subgenomic RNAs (HCVr). As expected, the NTP/dNTP ratio is elevated in HCVr cells. Compared with that of the control Huh7 cells with sh-scramble, the NTP/dNTP ratio of the RRM-knockdown cells is elevated. Knockdown of RRM1 or RRM2 increases HCV replication in HCV replicon cells. Moreover, inhibitors to RRMs, including Didox, Trimidox and hydroxyurea, enhance HCV replication. Among various HCV viral proteins, the NS5A and/or NS3/4A proteins suppress the expression of RRMs. When these are taken together, the results suggest that HCV down-regulates the expression of RRMs in cultured cells to promote its replication.

## 1. Introduction

Over 50% of the patients infected with hepatitis C virus (HCV) develop chronic infection. These chronic hepatitis C (CHC) patients are at high risk to develop liver cirrhosis and even hepatocellular carcinoma [1]. HCV pathogenesis is a very complex phenomenon and has not been fully understood. CHC patients can now be cured using various direct-acting antivirals (DAAs) [2]. Thus, the prevalence of HCV has fallen dramatically since oral DAAs were introduced. However, the global prevalence of CHC remains as high as 58 million people according to the estimation from the World Health Organization [https://www.who.int/news-room/fact-sheets/detail/hepatitis-c (assessed on 10 May 2023)]. Thus, HCV infection still poses a serious public health threat worldwide.

HCV, containing a single-stranded RNA genome with positive-polarity, belongs to the family Flaviviridae and genus hepacivirus. HCV relies highly on its host cells to propagate successfully. For example, host cellular factors, including coding and noncoding genes, have been identified to be dysregulated during HCV infection to favor viral replication [3].

Ribonucleotide reductase (RR; RNR) transforms RNA building blocks to DNA building blocks by catalyzing the substitution of the 2′OH-group of a ribose with a hydrogen by a mechanism involving protein radicals [4,5]. Mammalian RRs are tetramers (α2 β2), which consist of two α subunits (ribonucleotide reductase M1; RRM1) and two β subunits (ribonucleotide reductase M2; RRM2) [6]. RRM1 contains the catalytic site and two different allosteric regulatory sites; RRM2 harbors a di-iron cofactor and a tyrosyl radical essential for RR activity. RR is critical for the regulation of DNA synthesis in cells, and it is encoded by all living organisms as well as large DNA viruses, such as the herpesviruses and poxviruses [7,8,9]. Moreover, human papillomavirus (HPV) and hepatitis B virus (HBV) have been reported to induce host cellular RRM2 expression to enhance viral replication [10,11,12].

The relationship between cellular RRs and RNA viruses (not including retroviruses whose replication is through a DNA intermediate) has not yet been established. It is reasonable to assume that RNA viruses increase cellular ribonucleotide content to benefit viral replication through suppression of either RRM1 or RRM2. To test the assumption, an RNA virus, HCV, was used in this study.

## 2. Materials and Methods

### 2.1. Chemical/Drugs

Recombinant interferon-α was used in this study and was purchased from Prospec-Tany Technogene Ltd. (Ness-Ziona, Israel). Chemicals, such as Didox (3,4-Dihydroxybenzohydroxamic acid) [13], Trimidox and hydroxyurea were obtained commercially (Cayman Chemical, Ann Arbor, MI, USA) [14,15].

### 2.2. Cell Culture and Viruses

Huh7 cells were cultured in Dulbecco’s modified Eagle’s medium (DMEM) and supplemented with 10% fetal bovine serum (FBS), 100 U/mL penicillin and 100 µg/mL streptomycin (Gibco, Billings, MT, USA). Huh7 cells with HCV subgenomic RNAs [16,17] were cultured in DMEM plus 10% FBS, 100 U/mL penicillin, 100 µg/mL streptomycin and 400 µg/mL G418. All cultured cells were maintained at 37 °C with 5% CO_2_.

Generation of infectious HCV particles (HCVcc) and infectivity assay in Huh7.5 cells were performed as described previously [16,18,19].

### 2.3. Quantitation of Intracellular Nucleoside Triphosphate (NTP) and Deoxynucleoside Triphosphate (dNTP) Pools

The intracellular NTP and dNTP pools were determined using LC-MS/MS described in a previous report [20]. Briefly, 1 × 10^8^ cells, after washed with PBS, were dissolved in 1 mL 60% methanol. After 20 s of vortexing, cells were frozen for 30 min at −20 °C. Then, cells were sonicated for 15 min in an ice bath. After filtration by 0.45 and 0.22 μm filters, the cell lysates were transferred into a new Eppendorf and dried by N2 (Centrifugal Concentrator, Vacuum). Then, samples were analyzed with LC-MS/MS (Thermo Scientific Accela LC Systems plus Thermo TSQ Quantum Ultra Triple Quadrupole LC Mass Spectrometer) following the manufacturer’s instructions.

### 2.4. RNA Extraction and Real-Time Reverse Transcriptase–Polymerase Chain Reaction (Real-Time RT-PCR)

Total RNAs were extracted from the cells using a TRIzol reagent (Invitrogen, Thermo Fisher Scientific, Waltham, MA, USA) following the manufacturer’s instructions. Next, the RNAs were converted into cDNAs using an oligo-dT or random hexamer as the primer. The High-Capacity cDNA Reverse Transcription Kit (Applied Biosystems, ThermoFisher Scientific, MA, USA) was used for the reverse transcription. Then, the LabStar SYBR qPCR Kit (TAIGEN Bioscience Corporation, Taipei, Taiwan) was used for the real-time PCR. Primers used for the real-time RT-PCR are listed in Table 1. β-actin mRNA was used as the internal control. Experiments were repeated at least three times and the MIQE guide was followed in performing real-time RT-PCR [21]. Data were analyzed using the Student *t* test, and *p* < 0.05 was considered statistically significant (* *p* < 0.05, ** *p* < 0.01, *** *p* < 0.001).

### 2.5. Plasmid Construction and DNA Transfection

The plasmids used in this study were constructed using standard protocols as described in our previous studies [16]. The PCR primers used for the cloning are listed in Table 1. DNA fragments of EGFP, different HCV genes and promoters of RRM1 and RRM2 were amplified from pEGFP-C1 plasmid, p90/HCV-FL plasmid or genomic DNA from Huh7 cells with Phusion™ High-Fidelity DNA Polymerase (Thermo Scientific, Waltham, MA, USA) and cloned into pcDNA3.1-V5-HisA, pLAS3w-pPuro or pGL3Basic vectors by digestion with restriction enzymes, followed by ligation with T4 DNA ligase (Takara Bio, San Jose, CA, USA). Expressing plasmids (pLAS3w-NS3-ha, pLAS3w-NS3/4A-ha, and pLAS3w-NS4b-ha) were from Dr. King-song Jeng as a gift. All plasmids used in this study were verified by sequencing. The polyethylenimine (PEI, linear, MW 25,000) used to transfect DNA into cells was purchased from Polysciences Inc. (Warrington, PA, USA).

### 2.6. Western Blotting Analysis

A Western blotting analysis was performed as described previously [22]. In brief, total proteins from cultured cells were collected and separated by SDS-PAGE and transferred to a 0.2 μm PVDF membrane (Pall corporation, Port Washington, NY, USA). The membrane was washed three times for 10 min with 1×PBST and hybridized with the individual primary antibodies. The primary antibodies used were antibodies against HCV NS5A protein (Meridian Life Science, Memphis, TN, USA), against HCV core and NS3 proteins (Abcam, Cambridge, MA, USA), against Myc tag clone 4A6 (Merck Millipore, Billerica, MA, USA), against V5 tag (Bio-Rad, Hercules, CA, USA), against RRM1, RRM2 and Beta-Actin (Genetex, Irvine, CA, USA) and against HA tag and GFP (Santa Cruz Biotechnology, Dallas, TX, USA). In this assay, β-actin was used as the loading control. Then, the membrane was washed three times for 10 min with 1×PBST and hybridized with the individual secondary antibodies at 1:4000 dilution in 1×PBST for 1 h at room temperature. The secondary antibodies used were goat anti-rabbit IgG (HRP), goat anti-mouse IgG (HRP) or goat anti-rabbit IgG (AP). The PVDF membrane was then washed three times with 1×PBST and enhanced chemiluminescence (ECL) reagent (Thermo Fisher Scientific, Waltham, MA, USA) was applied. Autography was obtained by X-ray film or UVP BioSpectrum 810 (Thermo Fisher Scientific, Waltham, MA, USA). The Western blotting results are representative of two or more experiments. After the assay, the relative amounts of different proteins were quantified using the software “Quantity One” (Bio-Rad, Hercules, CA, USA).

### 2.7. The shRNA Knockdown and Stably Over-Expressed Experiments

The shRNA knockdown reagents were obtained from the National RNAi Core Facility in the Institute of Molecular Biology/Genomic Research Center, Academia Sinica, Taiwan. The shRNA knockdown (e.g., sh-RRM1, sh-RRM2) and stably over-expressed (e.g., HCV NS5A) experiments were performed with the lentiviral expressing system (http://rnai.genmed.sinica.edu.tw accessed on 15 April 2023), following the manufacturer’s instructions and our previous procedures [16]. In brief, pseudo-typed lentiviruses were generated by co-transfecting HEK-293T cells with plasmid DNA mixtures of pCMVΔ8.91, pMD.G and the plasmid for shRNA knockdown or over-expressed gens using a PEI transfection reagent. The pseudo-typed lentiviruses were harvested at 48 h post transfection. After harvest, the lentiviruses were filtered through a 0.45 μm filter, and used for transduction with 8 μg/mL polybrene, followed by puromycin (2 μg/mL) selection in Huh7 cells.

### 2.8. Luciferase Assay

Huh7 cells were cultured in DMEM with 10% FBS. 1 × 10^5^ cells were transfected with the reporter pGL3Basic/pGL3-RRM1P, the internal control plasmid pRL-TK (Promega, Madison, WI, USA) and different expressing plasmids. Cells were harvested 48 h after transfection. The dual-luciferase assay system (Promega, Madison, WI, USA) was used following the manufacturer’s instructions and our previous procedures [16]. In each experiment, triplicate samples were analyzed. The results shown were the average of three different experiments. ANOVA was used to compare the means among groups. A post hoc Tukey test was used to compare the means between two groups.

## 3. Results

### 3.1. HCV Suppressed the Expression of RRM1 and RRM2 and Up-Regulated the Intracellular NTP Level

To test the hypothesis that RNA viruses may reduce the expression of cellular ribonucleotide reductase (either RRM1, RRM2 or both), we first asked whether HCV suppressed the expression of RRM1 and/or RRM2. The expression of both RRM1 and RRM2 was down-regulated by HCV infectious virions (Figure 1). Further, the expression of both RRM1 and RRM2 was also suppressed in HCV replicon cells (HCVr, Huh7 with HCV subgenomic RNA) but not in the parental Huh7 cells (Figure 2A, no interferon-alpha (INF-α) treatment). Thus, HCV infection suppressed the expression of both ribonucleotide reductase subunits, RRM1 and RRM2. The mRNA levels of RRM1 and RRM2 were both reduced in the HCVr cells as well, compared with those in Huh7 cells (Figure 2B). Thus, HCV down-regulates the expression of RRM1 and RRM2, possibly at the transcriptional level. After removal of HCV subgenomic RNA by INF-α treatment (Figure 2C), the expression of RRM1 and RRM2 in HCVr cells increased (Figure 2A, right panel), indicating the expression of RRM1 and RRM2 is modulated by HCV. However, it is interesting to note that, INF-α treatment decreased the expression of RRM1 to 78% and RRM2 to 60% in Huh7 cells (Figure 2A, left panel).

The intracellular NTP and dNTP pools were measured in Huh 7 and HCVr cells (Table 2). ATP and dGTP were inseparable in this assay [20]. In previous reports, the amount of ATP has been reported to be much more than that of dGTP within cells [23,24]; thus, the amount of dGTP in the pool of ATP/dGTP is negligible. In sum, the intracellular total NTP level in HCVr cells was higher than that of Huh7 cells (Table 2).

### 3.2. Reduced Expression of RRMs Led to a Higher Intracellular NTP/dNTP Ratio

Knockdown of RRM1 or RRM2 individually by different shRNAs showed a significant reduction in expression of these genes in Huh7 cells (Figure 3). The intracellular NTP/dNTP amount in these cells was also measured in one clone of sh-RRM1 cells (Table 3) and one clone of sh-RRM2 cells (Table 4). Overall, the NTP amount in sh-RRM1 or sh-RRM2 cells were elevated, compared with that of the control (sh-scramble) cells (Table 3 and Table 4).

### 3.3. Reduced Expression of RRMs Led to Enhancement of HCV Replication

To determine the effect of RRMs on the HCV replication, HCVr cells with different shRNAs targeting RRMs were established separately (Figure 4A). Knockdown of RRMs by shRNAs showed a significant reduction in RRM expression and enhancement of HCV replication at both the protein (Figure 4A) and genomic RNA (Figure 4B) level.

### 3.4. Inhibition of RRMs’ Activity Facilitated HCV Replication

To further verify the effect of RRMs on HCV replication, various RRM inhibitors were used. Didox, a synthetic antioxidant, is one of the most potent ribonucleotide reductase inhibitors developed to date [13,15]. Didox treatment enhanced HCV replication dose-dependently in HCVr cells as shown at both the protein (Figure 5A) and genomic RNA (Figure 5B) levels. Trimidox treatment also enhanced the HCV replication dose-dependently in HCVr cells (Figure 5C). Moreover, hydroxyurea treatment facilitated the HCV replication dose-dependently in HCVr cells (Figure 5D). Thus, HCV replication is enhanced when RRM activity is inhibited.

### 3.5. RRM Expression Was Suppressed by HCV Viral Proteins NS5A and/or NS3/4A

To determine which viral gene(s) suppresses the expression of RRMs, different approaches were conducted. We first compared RRM1 and RRM2 expression in Huh7 cells with the controlled protein GFP and, in cells, it stably transfected with the HCV NS5A protein. Both RRM1 and RRM2 protein expression levels were lower in cells with NS5A protein than those in cells with GFP (Figure 6A). Furthermore, the mRNA levels of RRM1 (Figure 6B) and RRM2 (Figure 6C) were suppressed by transiently transfected NS5A dose-dependently, while the mRNA levels of RRM1 (Appendix A) and RRM2 (Appendix A) were not affected by transiently transfected NS4B.

To further confirm that NS5A down-regulates the expression of RRM1 and RRM2 at the transcriptional level, the promoter regions of RRM1 (−1000 to +300 of RRM1 gene) or RRM2 (−1510 to +50 of RRM2 gene) were constructed in front of a luciferase reporter gene, individually [25,26,27,28]. Indeed, the promoter activities of RRM1 and RRM2 were suppressed by transiently transfected NS5A dose-dependently (Figure 6D), while the promoter activities of RRM1 and RRM2 were not affected by transiently transfected NS4B (Appendix A).

In addition, transiently transfected HCV NS3/4A reduced the expression of RRM2 but not RRM1, dose-dependently, at the protein level (Figure 7A). As expected, the promoter activity of RRM2 was also suppressed by transiently transfected NS3/4A, dose-dependently (Figure 7B).

The HCV core protein regulates the expression of many cellular genes [29]. To determine whether core protein would down-regulate the expression of RRM1 and RRM2, various amounts of plasmids expressing core protein were transfected into Huh7 cells. Under these conditions, neither the expression of RRM1 nor that of RRM2 was affected by the core protein (Appendix A).

### 3.6. Transcriptomic Data from NCBI GEO Database

To determine whether HCV really affects the expression of RRM1 and/or RRM2, the RNA sequence data from NCBI GEO database (accession GSE166428), including Huh-7.5 cells and primary human hepatocytes (PHH), (infected by HCV or not), were analyzed [30]. In contrast with that of mock-infection, the expression of RRM1 and RRM2 decreased 6 h after Huh-7.5 cells were infected with HCV (Appendix A). The expression of RRM2 also decreased, while the expression of RRM1 increased 72 h after Huh-7.5 cells were infected with HCV (Appendix A). In contrast with that of mock-infection, the expression of RRM1 and RRM2 decreased 6 h after PHH cells were infected with HCV (Appendix A). The expression of RRM1 also decreased, while the expression of RRM2 increased 72 h after PHH cells were infected with HCV (Appendix A). Furthermore, the RNA sequence data from diagnostic liver biopsies of patients with or without CHC were also analyzed (accession GSE84346 at NCBI GEO database) [31]. In contrast with that of normal liver biopsies, the expression of RRM1 and RRM2 increased in liver biopsies with CHC (Appendix A).

## 4. Discussion

In this study, down-regulation of RRM expression was detected in HCV infectious virions (Figure 1) and HCVr cells (Figure 2). Moreover, the intracellular NTP level in HCVr cells is higher than that of Huh7 cells (Table 2). Thus, HCV suppressed the expression of RRM1 and RRM2 and up-regulated the intracellular NTP level. On the other hand, a reduction in the expression of RRMs by shRNAs could increase the intracellular NTP/dNTP ratio of these cells (Figure 3, Table 3 and Table 4). Additionally, knockdown of RRMs showed a significant enhancement in HCV replication (Figure 4). In addition, three different inhibitors to RRMs facilitated HCV replication (Figure 5). In summary, our results suggest that suppression of RRMs (by HCV, shRNAs or inhibitors) did enhance HCV replication. Reduced expression of RRM1 and RRM2 was also detected for 6h after Huh 7.5 and PHH cells were infected with HCV (Appendix A).

Though the total NTP amount in HCVr cells is higher than that in Huh7 cells, it is interesting to notice that the amount of UTP and GTP were slightly more reduced in HCVr than that in Huh7 cells (Table 2). This may have been caused by other unspecified supply sources compensating for the downregulation of RRMs in HCVr cells. Alternatively, it could be because HCV RNA replication is active in HCVr cells, which possibly consumed the UTP and GTP in the host cells. Further studies are needed to clarify this issue. It is also worth mentioning that not only the dNTP amount but also the NTP amount was reduced when RRM1 or RRM2 was knocked-down in the H23 non-small cell lung cancer cells [24]. This may indicate a dNTP/NTP ratio, but not an individual amount of NTP or dNTP is important.

Herpesviruses express their own viral ribonucleotide reductases [9]. HBV and HPV induce host cellular RRM2 expression for viral replication [10,11,12]. Not surprisingly, previous reports have also demonstrated that RR inhibitors (Didox, Trimidox and/or hydroxyurea) could be used as antiviral agents against herpes, cytomegalovirus and murine acquired immunodeficiency syndrome [32,33]. Moreover, knockdown of RRM2 inhibited HBV replication and RRM2 inhibitors (e.g., Pterostilbene) suppressed HBV replication and hepatocellular carcinoma proliferation [34,35]. Therefore, RR inhibitors should be able to suppress the replication of DNA viruses while enhancing that of RNA viruses.

The expression of RRMs was reduced in HCVr cells compared to that of parental Huh7 cells (Figure 2). In the HCVr cells, only the viral polyprotein containing NS3/4A-NS4B-NS5A-NS5B is encoded [17]. Thus, the viral proteins (NS3/4A, NS4B, NS5A and/or NS5B) expressed in HCVr cells should be involved in the suppression of RRM expression. However, the role of each viral protein in modulation of RRM expression is not clear [36]. Unlike NS4B, NS5A down-regulated both RRM1 and RRM2 expression at the transcriptional level (Figure 6). In addition, NS3/4A suppressed RRM2 but not RRM1 at the transcriptional level (Figure 7). Expression of NS5B could not be detected in either transiently or stably transfected assays, which may be due to the lability of this protein [36]. Thus, HCV suppresses RRM expression, possibly through the viral proteins NS5A and NS3/4A at the transcriptional level.

A recent report has shown that no significant difference was observed in dCTP levels between HCV-infected JFH/K4 cells and uninfected HuH-7 cells [37]. In agreement with this finding, dCTP levels between Huh7 and HCVr are similar in our present study (Table 2). In this previous report, Kitab et al. demonstrated that RRM2 is a cellular factor essential for HCV replication. Mechanically, RRM2 promotes HCV RNA replication by protecting NS5B protein from hPLIC1-dependent proteasomal degradation. It showed that increased RRM2 mRNA and protein expression levels were detected in HCV-infected hepatocytes from chimeric mice and also in hepatoma cells infected with the HCV strain JFH1 [37]. The cause of discrepancies between our study and Kitab et al. [37] is unclear. They may be the result of a mechanism that allows HCV to survive in different infection conditions. We assume the intracellular NTP levels (or NTP/dNTP ratio) are high in the cells used in this previous report. When the intracellular NTP level (or NTP/dNTP ratio) is high enough for HCV replication, i.e., expression of RRM2 is low, RRM2 could promote HCV replication through protecting the NS5B protein from degradation. On the other hand, when the intracellular NTP level (or NTP/dNTP ratio) is low, i.e., expression of RRM2 is high, HCV would suppress the expression of RRMs through NS5A and/or NS3/4A to increase the cellular NTP level (or NTP/dNTP ratio). Further studies to uncover the detailed molecular mechanisms regarding how HCV modulates RRM2 expression are needed to clarify this issue. This assumption is supported by the PHH cells infected with HCV. The expression of RRM1 decreased at both 6 h and 72 h of PHH after infection with HCV, while the expression of RRM2 decreased at 6 hrs but increased at 72 h of PHH after infection with HCV (Appendix A).

Our present results were not supported by the transcriptomic data from diagnostic liver biopsies of patients with or without CHC (Appendix A), indicating that HCV replication in cultured cells may not fully reflect that in CHC patients. On the other hand, reduced expression of RRM1 and RRM2 was also detected for 6 h after HCV infection of Huh7.5 and PHH cells (Appendix A), suggesting that our present study may reflect the initial HCV infection of hepatocytes. More studies are needed to support this hypothesis.

## 5. Conclusions

In summary, results in this study have suggested that HCV suppresses the expression of RRMs to increase the intracellular NTP level, and in turn to promote viral replication in cultured cells.

## Figures and Tables

**Figure 1 pathogens-12-00892-f001:**
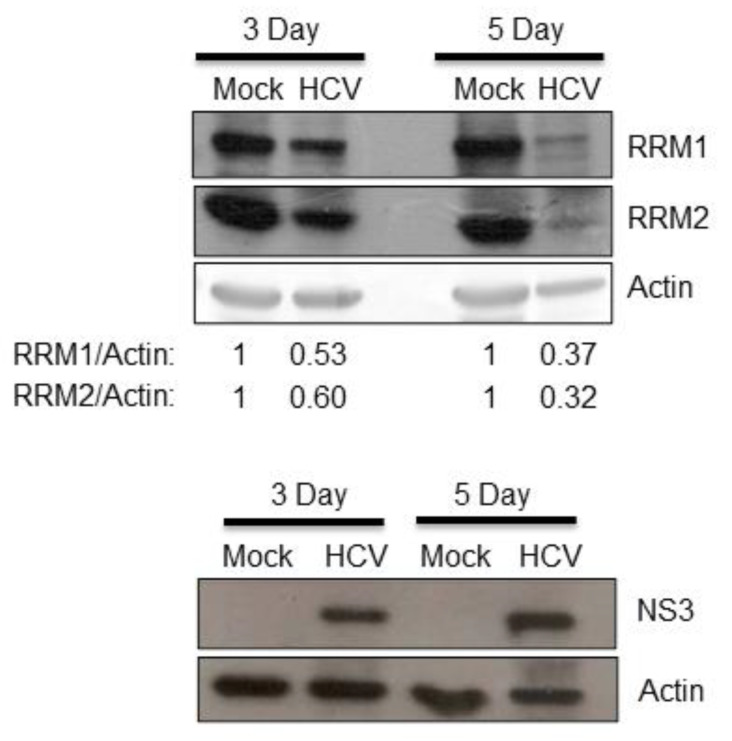
Huh 7.5 cells were either mock-infected or infected with infectious HCV (M.O.I. = 0.5). Three or five days after infection, protein samples derived from these cells were analyzed by Western blotting against RRM1 or RRM2. The presence of HCV in the cells was demonstrated by HCV NS3 protein expression.

**Figure 2 pathogens-12-00892-f002:**
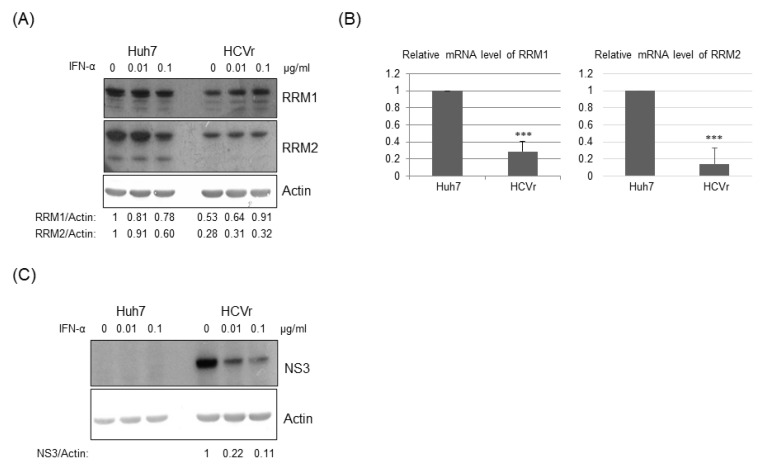
(**A**) Western blotting analysis of the expression of RRM1 and RRM2 in Huh7 cells and HCV replicon cells with or without interferon treatment. (**B**) The mRNA levels of RRM1 and RRM2 in Huh7 cells and HCV replicon cells were determined by real-time RT-PCR. (**C**) Western blotting analysis demonstrated that the HCV subgenomic RNA level is inversely proportionate to the dosage of interferon-alpha used. The HCV NS3 protein expression level reflects the amount of HCV subgenomic RNAs, and actin was applied as a loading control. *** *p* < 0.001.

**Figure 3 pathogens-12-00892-f003:**
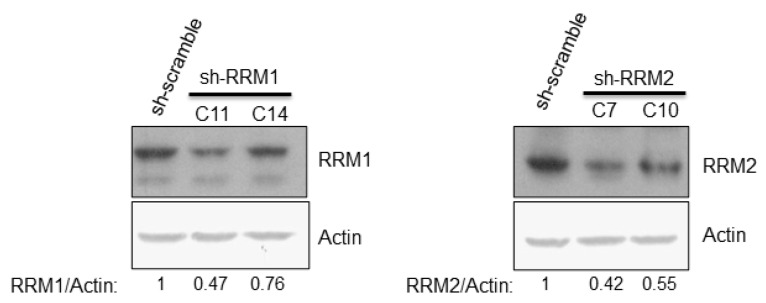
Western botting analysis showed the protein expression level of RRM1 and RRM2 after shRNA knockdown in the Huh7 cells.

**Figure 4 pathogens-12-00892-f004:**
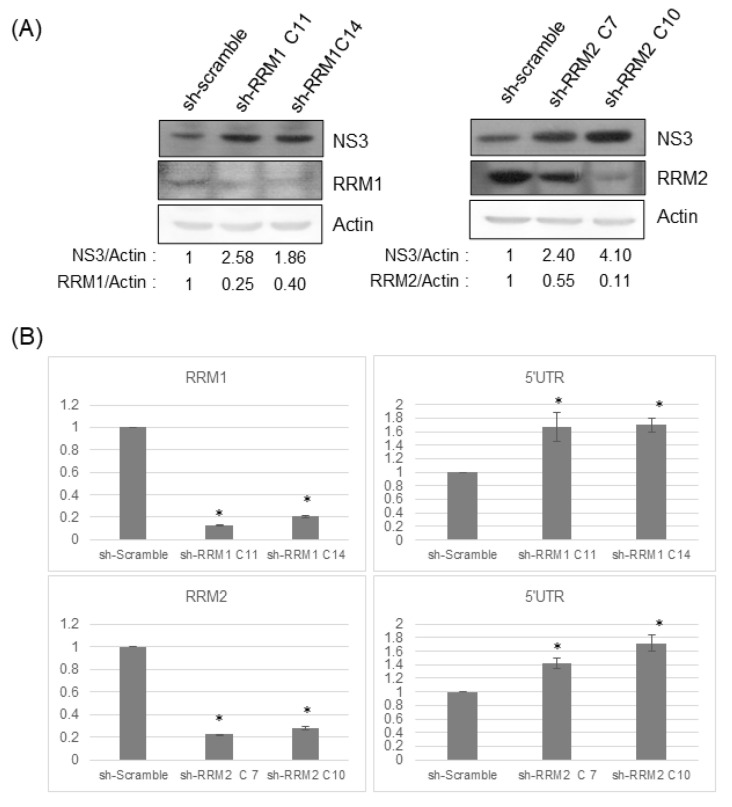
(**A**) Knockdown of either RRM1 or RRM2 enhances HCV replication. Western blotting analysis of the protein expression in HCV replicon cells stably transfected with control sh-scramble or various shRNA clones targeting RRM1 or RRM2. (**B**) The amount of RRM1 or RRM2 mRNAs (left panels) or the amount of HCV subgenomic RNAs (right panels) determined by real-time RT-PCR in HCV replicon cells with various shRNA clones. * *p* < 0.05, ** *p* < 0.01, *** *p* < 0.001.

**Figure 5 pathogens-12-00892-f005:**
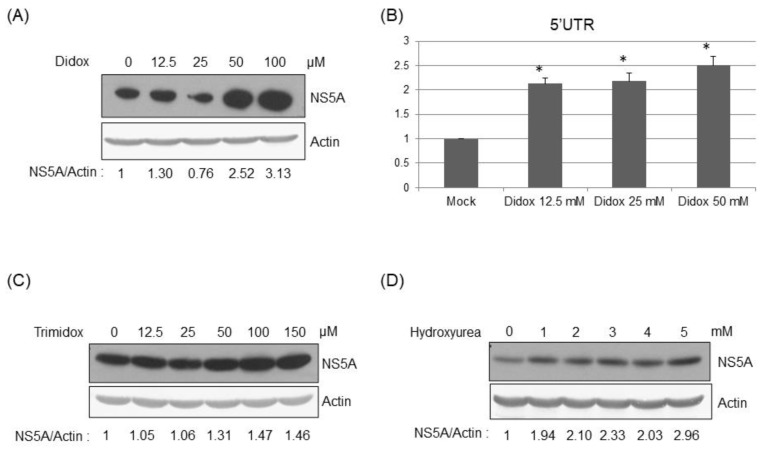
Treatment of RRM inhibitors enhanced hepatitis C viral replication. Western blotting analysis of the HCV NS5A protein to reflect the viral replication in the HCVr cells 48 h after the treatment of various amounts of Didox (**A**), Trimidox (**C**), or hydroxyurea (**D**). (**B**) Real-time RT-PCR analysis of HCV 5′-UTR to reflect viral RNA amount in HCVr cells 48 hrs after the treatment of various amounts of Didox. * *p* < 0.05.

**Figure 6 pathogens-12-00892-f006:**
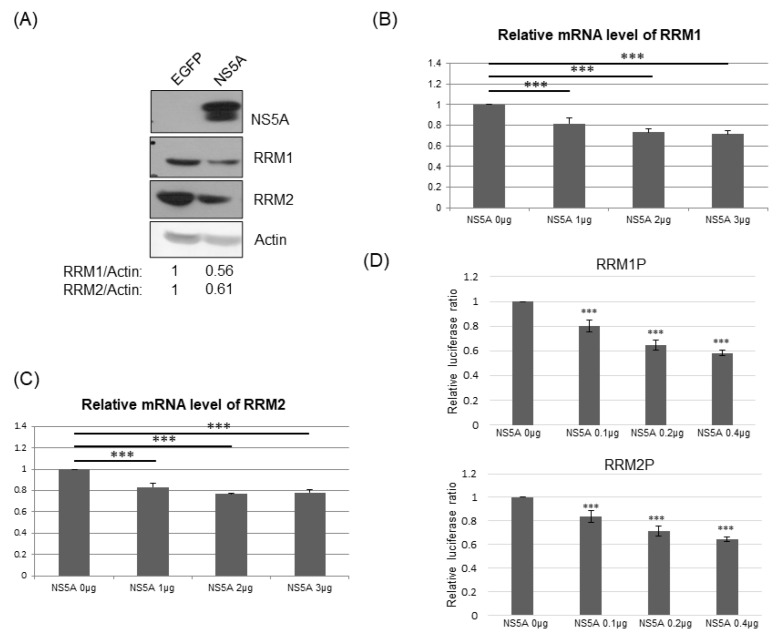
(**A**) Western blotting analysis of RRM1 or RRM2 protein expression in Huh7 cells stably transfected with control EGFP or HCV NS5A protein. (**B**,**C**) Real-time RT-PCR analysis of the mRNA expression of RRM1 (**B**) or RRM2 (**C**) 48 hrs after the transfection of various amounts of vectors and/or plasmids expressing NS5A protein into Huh7 cells. (**D**) Various luciferase reporter assays were performed 48 h after transfection of reporters for promoter activity of RRM1 (upper) or RRM2 (lower) and various amounts of vectors and/or plasmids expressing NS5A protein into Huh7 cells. *** *p* < 0.001.

**Figure 7 pathogens-12-00892-f007:**
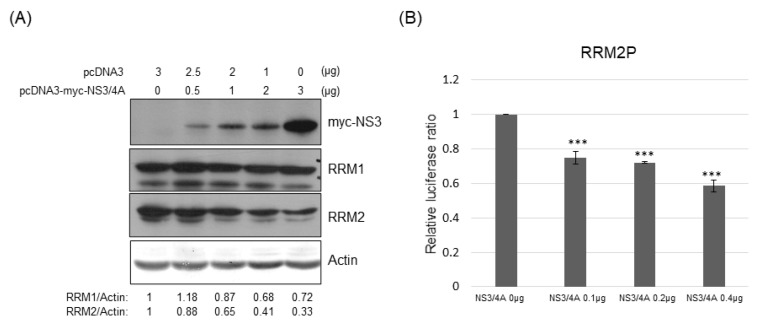
(**A**) Western blotting analysis of RM1 or RRM2 in Huh7 cells transfected with various plasmids as indicated; 48 h after transfection, protein samples derived from these cells were analyzed with myc tag, RRM1 or RRM2. (**B**) Various luciferase reporter assays were performed 48 h after transfection of reporters for promoter activity of RRM2 and various amounts of vectors and/or plasmids expressing NS3/4A protein into Huh7 cells. *** *p* < 0.001.

**Table 1 pathogens-12-00892-t001:** Primer sequences used in this study.

Primer Sequences for the Cloning of Expressing Plasmids	
EGFP-S2	5′- CTAGCTAGCATGGTGAGCAAGGGCGAGGA-3′
EGFP-AS3	5′- GCTCTAGACTTGTACAGCTCGTCCAT-3′
Core-S	5′-CGGAATTCATGAGCACGAATCCTAA-3′
Core-AS3	5′GCTCTAGAGGCTGAAGCGGGCACAGT-3′
NS3/4A-S	5′-CGGGATCCGCGCCCATCACGGCG -3′
NS3/4A-AS	5′-GCTCTAGACTATTAGCACTCTTCCATCTC -3′
NS4B-F	5′-CGGAATTCATGTCTCAGCACTTACCGTAC -3′
NS4B-AS	5′-GCTCTAGATTAGCATGGAGTGGTACA-3′
NS5A-S	5′-CGGAATTCATGTCCGGTTCCTGGCTAAG -3′
NS5A-AS3	5′-GCTCTAGAGCAGCACACGACATCTTC -3′
RRM1P1000-S	5′-GGGGTACCACCATGCCTGGCTACT-3′
RRM1P+300-AS	5′-GAAGATCTATCCAAGACTGGACTGCG-3′
RRM2P1510-S	5′-CTAGCTAGCTTCCTGGAGATGGATGCTTTA-3′
RRM2P+50-AS	5′-GAAGATCTTGGCTGCGCCTTGC-3′
**Primer Sequences for the Real-Time RT-PCR Analysis**	
QRRM1-F	5′-ACCGCCCACAACTTTCTAG-3′
QRRM1-R	5′- CCAGTAGCCCGAATACAACTC-3′
QRRM2-F	5′- AAGGACATTCAGCACTGGG-3′
QRRM2-R	5′- AGCGGGCTTCTGTAATCTG-3′
Qbetaactin-F	5′-CATCGAGCACGGCATCGTCA-3′
Qbetaactin-R	5′-TAGCACAGCCTGGATAGCAAC-3′
HCV-RTF	5′-AGCGTCTAGCCATGGCGT-3′
HCV-RTR	5′-CAAGCACCCTATCAGGCAGT-3′

**Table 2 pathogens-12-00892-t002:** Nucleoside triphosphate levels (uM) determined by LC-MS/MS in Huh7 and HCV replicon cells.

	Huh7	HCVr	*p*-Value
ATP/dGTP	356.9 ± 5.8	421.2 ± 10.7	0.0003
UTP	181.5 ± 12.5	179.6 ± 4.4	0.4068
CTP	67.4 ± 3.1	82.1 ± 8.4	0.0238
GTP	252.1 ± 5.9	233 ± 8.4	0.0161
dATP	2.8 ± 0.5	6.8 ± 0.6	0.0004
dTTP	11.5 ± 2.7	9.7 ± 2.4	0.2180
dCTP	1.4 ± 0	1.7 ± 0.5	0.1879

**Table 3 pathogens-12-00892-t003:** Nucleoside triphosphate levels (uM) determined by LC-MS/MS in Huh7 with control sh-scramble or with shRRM1 C11.

	sh-Scramble	sh-RRM1 C11	*p*-Value
ATP/dGTP	354.2 ± 14.1	531.5 ± 59.8	0.003754
UTP	145.8 ± 13.4	247.4 ± 36.2	0.005197
CTP	41.8 ± 4.7	21.3 ± 2	0.00113
GTP	165.6 ± 4.9	281.1 ± 34	0.002163
dATP	4.8 ± 1	2.1 ± 0.3	0.006417
dTTP	8.9 ± 1.8	16.6 ± 1.6	0.002676
dCTP	1.7 ± 0.3	0.6 ± 0.2	0.003065

**Table 4 pathogens-12-00892-t004:** Nucleoside triphosphate levels (uM) determined by LC-MS/MS in Huh7 with control sh-scramble or with shRRM2 C10.

	sh-scramble	sh-RRM2 C10	*p*-Value
ATP/dGTP	354.2 ± 14.1	714.9 ± 118.5	0.003178
UTP	145.8 ± 13.4	354.8 ± 108.2	0.014691
CTP	41.8 ± 4.7	25.7 ± 4.2	0.005665
GTP	165.6 ± 4.9	598.6 ± 35.9	0.000016
dATP	4.8 ± 1	3.1 ± 0.5	0.032387
dTTP	8.9 ± 1.8	5.7 ± 0.8	0.02581
dCTP	1.7 ± 0.3	1.7 ± 0.4	0.423585

## Data Availability

Not applicable.

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
