# Peer review of "Hepatitis C Virus Down-Regulates the Expression of Ribonucleotide Reductases to Promote Its Replication"

_pathogens, 2023, doi:10.3390/pathogens12070892_

Round 1

Reviewer 1 Report

Chronic Hepatitis C virus (HCV) infection is a global health issue that poses serious threat to public health. As the pathogenesis of HCV is complex and not completely understood, delineating mechanisms by which HCV promotes its replication in the human host is important.

In this study, the authors aim to unearth a possible mechanism that promotes HCV replication in humans. I would like to suggest the following revisions:

1.      Figure 2B, statistical tests were performed to determine whether differences in expression level are significant.

2.      Lines 165-166: “The intracellular NTP level in HCVr cells is much higher than that of 165 HuH7 cells”. Although there is an increase in the level of some NTPs (GTP and ATP) there is a marginal increase in the levels of other NTPs. Therefore, it is not correct to say NTP levels were much higher.

3.      The discussion section of the paper is extremely small, the authors must explain in much more detail why they specifically studied the structural proteins (NS5A, NS3/4 etc.)

4.      Line 343-344: “It may be a mechanism for 343 HCV to survive in different infection conditions”. What are the differences in infection conditions used in this study and the previous study. This should be discussed in detail.

5.      In method section, the authors must briefly describe the different methods (3-4 lines summarizing the methods). Just referring to methods used in previous studies is not sufficient.

Significant editing for certain sections is required 

Reviewer 2 Report

 Yang and colleagues analyze in their interesting study entitled ‘Hepatitis C virus down-regulates the expression of ribonucleo-2 tide reductases to promote its replication’ the effects on HCV on ribonucleo-2 tide reductases (RRM1 and RRM2) and vice versa. Using two well established systems (Huh7.5 cells for HCV infection and Huh7 cells for HCV sub-genomic replicon experiments) the authors observe a RRM1 and 2 downregulation on both mRNA and protein level after HCV infection and sub-genomic driven HCV replication. Vice versa, overexpression studies of RRM1 and RRM2 show a downregulation of HCV replication. Further, the authors transiently transfect individual HCV proteins and RRM promoter regions to decipher the involved viral proteins and the mode of inhibition.

Although the reported effects seem to be in a low to intermediate change (e.g. ~2 fold on HCV replication/ protein expression) they might provide an interesting insight into the interplay between HCV and the host cell.

Thus said, there are several technical and conceptual issues which should be considered to improve the overall quality of the present manuscript:

1)      As hepatocytes are usually quiescent and non-replicating, RRM levels are expected to be low. Hence, expression levels can be very different between Huh7(.5) cells and hepatocytes within a liver. Therefore, I suggest that the authors collect transcriptomic data from several studies in which either primary human hepatocytes (PHHs) +/- HCV was analyzed or liver tissue +/- HCV as analyzed. I recommend to mine several studies and to present the results not normalized between each other, so the reader has a chance to see the RRM variation within non-infected PHHs / liver tissue and in comparison to HCV infected tissue/ PHHs. Several studies are out there, just two examples: GSE166428 (https://www.ncbi.nlm.nih.gov/geo/query/acc.cgi?acc=GSE166428 ) or GSE84346 (https://www.ncbi.nlm.nih.gov/geo/query/acc.cgi).

Looking and this transcriptomic data from a more physiological setting will show whether HCV infection really affects RRM expression or whether the authors are chasing a Huh7 specific phenotype which does not have much implications on HCV. If true, any other RNA virus replicating in cells with higher RRM levels would be a better prototype to study RNA virus-RRM interactions than HCV.

But if transcriptomic data are in line with the presented results gained from Huh7 cells, this would highly support the findings and their relevance and increase the overall message of the manuscript.

2)      Although the single experiments seem to be solid, the authors don’t state how often individual experiments were performed independently (not technical replicates but independent experiments). This is especially the case for qRT-PCR images and luciferase assays where error bars are shown (several figures). Further, no description exists in the method section which statistical tests have been applied. Also, the value range of the asterisks in the figures is not explained.

Minor comments:

Please provide a reference for the statement that ATP is much more abundant than dGTP.

Thea article can be read fluently, language was clear to a non-native english speaker.

Reviewer 3 Report

In this manuscript, the authors demonstrated that HCV, in particular NS5A and NS3/4A, silences ribonucleotide reductases (RRMs) to support viral replication by expanding host NTP pool. These findings are interesting and the manuscript, overall, is well written. I, nonetheless, have following concerns that the authors should address.

1.      Figure 2(A): The band intensities should be integrated into those of relative values to parental HuH7 cells without IFNα treatment.

2.      Table 2: Although the authors concluded that "the intracellular NTP level in HCVr cells is much higher than that of HuH7 cells" (lines 165-166), the data insufficiently support this. Indeed, UTP and GTP were reduced. This critical issue must be addressed. The followings are examples:

a.       They can be produced not only by de novo synthesis, but also by other mechanisms including salvage pathway and oxidative phosphorylation. Thus, is it possible that such other supply sources compensate the downregulation of RRMs? 

b.       In HCVr cells, because HCV replication is active, they possibly consume NTP pools in host cells. This measurement should be done in the presence of an NS5B polymerase inhibitor, such as sofosbuvir.

c.       RRM1/2 synthesize dNDP from NDP. The authors should determine intracellular NDPs and dNDP levels by LC-Ms/Ms.

Round 2

Reviewer 1 Report

The authors have satisfied all major concerns. The manuscript can be accepted in the current form. 

Minor editing is required. 

Reviewer 2 Report

Thank you very much for providing a prompt revision for the study. All my concerns have been sufficiently addressed.

The paper reads fluent for a non-native speaker.

Reviewer 3 Report

Although the authors have well addressed my comments, one minor concern remains to be solved.
Figure 2A: It is confusing that there are many values for one band. The first 4 lines from the top should be removed, leaving the following 2 lines.

   RRM1/Actin  1  0.81  0.78      0.53  0.64  0.91
  RRM2/Actin  1  0.91  0.60      0.28  0.31  0.32
